

# Artificial light pollution increases nocturnal vigilance in peahens

Jessica L. Yorzinski[1,2], Sarah Chisholm[3], Sydney D Byerley[2], Jeanee R. Coy[2], Aisyah Aziz[2], Jamie A. Wolf[1] and Amanda C. Gnerlich[2]

[1] Department of Biological Sciences, Purdue University, West Lafayette, IN, United States
[2] Department of Animal Sciences, Purdue University, West Lafayette, IN, United States
[3] Centre of Computational Statistics and Machine Learning, University College London, London, United Kingdom

## ABSTRACT

Artificial light pollution is drastically changing the sensory environments of animals. Even though many animals are now living in these changed environments, the effect light pollution has on animal behavior is poorly understood. We investigated the effect of light pollution on nocturnal vigilance in peahens (*Pavo cristatus*). Captive peahens were exposed to either artificial lighting or natural lighting at night. We employed a novel method to record their vigilance behavior by attaching accelerometers to their heads and continuously monitoring their large head movements. We found that light pollution significantly increases nocturnal vigilance in peahens. Furthermore, the birds faced a trade-off between vigilance and sleep at night: peahens that were more vigilant spent less time sleeping. Given the choice, peahens preferred to roost away from high levels of artificial lighting but showed no preference for roosting without artificial lighting or with low levels of artificial lighting. Our study demonstrates that light pollution can have a substantial impact on animal behavior that can potentially result in fitness consequences.

## INTRODUCTION

Humans are rapidly altering natural environments and this can lead to dramatic changes in the sensory landscape. One change to the sensory landscape that has particularly pronounced effects on wildlife is artificial light (*Longcore & Rich, 2004*; *Tuomainen & Candolin, 2011*; *Sol, Lapiedra & González-Lagos, 2013*; *Gaston, Duffy & Gaston, 2014*). Artificial light is created by many different sources, such as streetlights, lighted buildings or towers, and security lights. Nearly 20% of land on earth is considered polluted by light (*Cinzano, Falchi & Elvidge, 2001*) and this pollution is increasing every year (*Hölker et al., 2010*). Light pollution has immediate fitness impacts on animals (*Rich & Longcore, 2006*). Animals that fail to adjust their behavior in response to artificial light can have reduced survival and reproductive success. In extreme cases, species may even become at risk of extinction (*Stockwell, 2003*).

Animals exhibit altered behavior in response to light pollution. Increased nocturnal illumination affects movement patterns. Rather than moving toward the sea, hatchling

Corresponding author
Jessica L. Yorzinski,
jyorzinski@purdue.edu

turtles are attracted to shoreline lights and fail to begin their oceanic migrations (*Tuxbury & Salmon, 2005*). The movement patterns of migrating birds are also disrupted. They are attracted to artificial lights on overcast nights and remain near those lights rather than continuing their migration (*Avery, Springer & Cassel, 1976*). Artificial light can impact courtship behavior. Songbirds initiate singing earlier in the morning and can even obtain more extra-pair mates when exposed to environments with artificial lighting (*Miller, 2006*; *Kempenaers et al., 2010*). In addition, light pollution can alter predator–prey interactions. Harbor seals are more successful at capturing salmonids in the presence of artificial light (*Yurk & Trites, 2000*). Birds and bats can likewise prey on moths at high rates when the moths congregate at artificial light sources (reviewed in *Frank, 1988*). Despite our growing knowledge on the effects of artificial light on animal behavior (*Rich & Longcore, 2006*; *Gaston, Duffy & Gaston, 2014*), we still know little about the mechanisms by which animals adjust their behavior in response to artificial nocturnal illumination (*Tuomainen & Candolin, 2011*; *Kurvers & Holker, 2015*).

In contrast, we do know that variation in natural lighting at night influences vigilance (*Beauchamp, 2007*). Depending on moon phase, light at night can vary between about 0.5 lux for a new moon and 2 lux for a full moon (*Weaver, 2011*). This variation alters vigilance levels differently depending on the species (*Beauchamp, 2015*). Greater flamingos and tammar wallabies increase their vigilance behavior at night when light levels are low (*Beauchamp & McNeil, 2003*; *Biebouw & Blumstein, 2003*) but gerbils decrease their vigilance behavior in response to low light levels (*Kotler et al., 2010*). Because nocturnal light levels can vary based on sleeping sites (*Gorenzel & Salmon, 1995*; *Longcore & Rich, 2007*), animals can choose to sleep under preferred lighting conditions (*Nersesian, Banks & McArthur, 2012*). Their choice of sleeping sites and vigilance behavior will in turn affect their sleep (*Gauthier-Clerc, Tamisier & Cézilly, 2000*). However, we do not know how prey species alter their nocturnal vigilance behavior when exposed to artificial lighting.

We therefore investigated the effects of light pollution on nocturnal vigilance behavior in peafowl. Peafowl are an appropriate species in which to examine this topic because they must increasingly live in well-lit urban environments due to habitat loss (*Ramesh & McGowan, 2009*). They are a lekking species that are native to the Indian subcontinent but have also been introduced to North America and other regions (*Kannan & James, 1998*). At night, they roost on tall structures (such as trees and poles; *De Silva, Santiapillai & Dissanayake, 1996*; *Parasharya, 1999*) and periodically open their eyes to scan their environment (*Yorzinski & Platt, 2012*). Many nocturnal predators, such as tigers, jackals, and raccoons, could potentially prey on them (*Harihar et al., 2007*; *De Silva, Santiapillai & Dissanayake, 1996*; *Kannan & James, 1998*).

We developed a novel method for monitoring vigilance rates by using accelerometers. Accelerometers have become an increasingly popular tool for studying animal behavior (e.g., *Sakamoto et al., 2009*; *Grünewälder et al., 2012*; *Nathan et al., 2012*). They are often attached to an animal's back and can be used to classify general activity patterns (e.g., flying, resting, walking; *Sakamoto et al., 2009*). Accelerometers that are attached to animals' heads can record head movements (*Kokubun et al., 2011*). Since high head

movement rates are related to heightened antipredator vigilance (e.g., *Jones, Krebs & Whittingham, 2007*), we can use head movement rates to approximate vigilance levels.

## METHODS

We examined the effect of artificial light pollution on vigilance levels in a captive population of adult peahens. The artificial light experiment was conducted between October 2013 and July 2014 at the Purdue Wildlife Area in West Lafayette, IN, USA (40.450327°N, −87.052574°E). The peafowl were housed in a large outdoor aviary (24.4 × 18.3 × 1.8 m) in an open area and were given food and water ad libitum. The study was approved by Duke University Animal Care and Use Committee (A205) and Purdue University Animal Care and Use Committee (1305000862 & 1504001232).

### Artificial light experimental procedure

We conducted thirteen light trials and thirteen control trials to test the effect of artificial light on vigilance behavior. A given bird was tested in either a light trial or a control trial (the order was randomized across birds; 26 different birds were therefore tested overall). For each trial, a female was transported to an experimental cage (9 m × 4.5 m). The experimental cage was a section within the main aviary that was surrounded by black plastic. The black plastic went from the ground to the roof on the two sides of the cage that faced the main aviary (this ensured that the trial bird was unable to see the birds in the flock) and from the ground to 1.15 m tall on the other two sides. It had a wooden sawhorse roost (0.85 m tall and 1.3 m long) that was positioned 4.5 m from an LED flood light (Philips 17-Watt Outdoor and Security Bright White; model: PAR38; flicker rate: 38 kHz; spectral radiance has two peaks: 4 mW/nm at 450 nm and 8.4 mW/nm at 600 nm (see Philips technical application guides for complete graph of spectral radiance)), which was suspended from the roof (1.8 m from the ground). Before the female was released into the experimental cage, a velcro strip (3.5 mm × 1.8 mm) with elastic straps was glued (Artiglio Super 620) to the feathers atop her head. After at least 1 h, a 3-axis accelerometer (TechnoSmart, Rome, Italy; 3 mm × 1.1 mm; 0.5 g; sample resolution: 19.6 m s$^{-2}$; sample rate: 50 Hz), which was protected in shrink wrap and electrical tape, was attached to the bird's head using velcro and secured by the strap (Fig. 1). The bird was then released into the experimental cage.

Each trial lasted seven nights. During a light trial, the light was off during nights 1, 6, and 7 and was on during nights 2–5 (this experimental design is similar to the one used in *Stone, Jones & Harris, 2009*). When the light was initially turned on during the daytime of the second trial day, it remained on (even during daylight) until the daytime after the fifth trial night. At night when the light was turned on, the light intensity was 1,260 lux below the light (light meter on ground facing up at light) and 0.75 lux at the roost (light meter facing toward the light); when the light was turned off, the light intensity was 0.04 lux below the light and 0.01 lux at the roost (Extech EasyView 31 light meter; resolution: 0.01 lux for readings below 20 lux and 1 lux for readings above 999; measurements taken during a night with clear skies and 69.5% moon illumination). During a control trial, the light was never turned on. An experimenter replaced the accelerometer each day of a

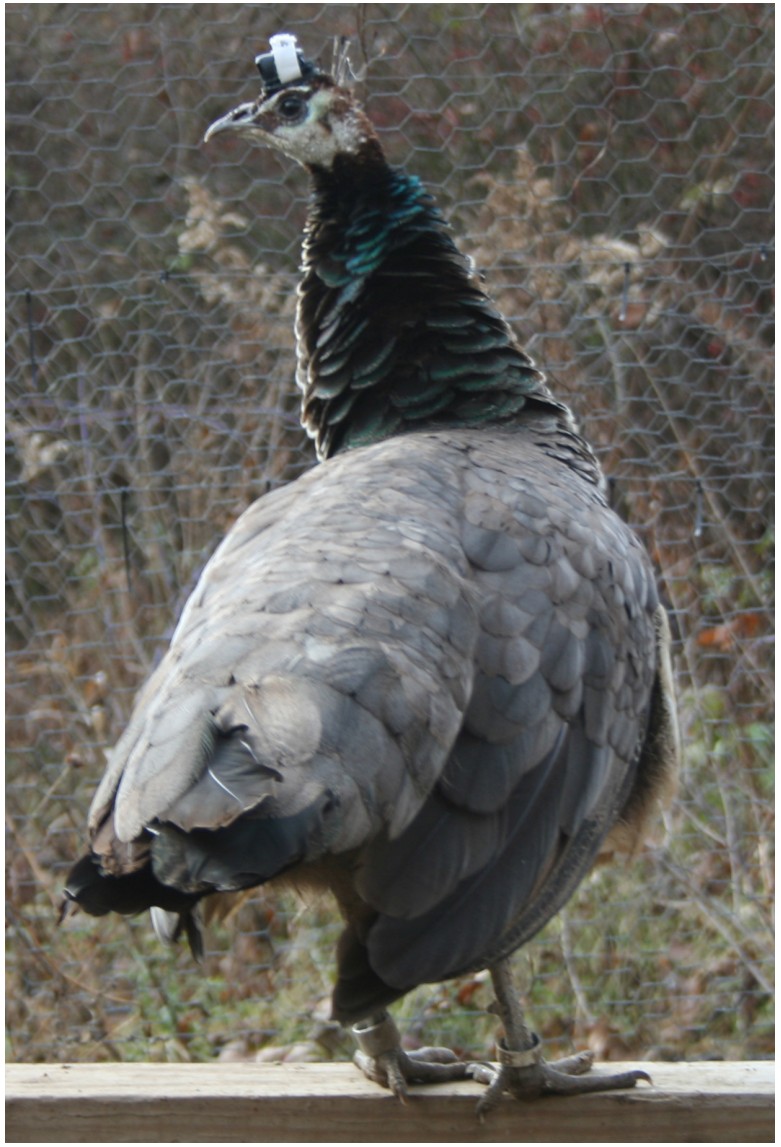

**Figure 1** Peahen on the roost wearing an accelerometer.

light and control trial (the accelerometer battery did not last more than 48 h) and did so at least 1 h after sunrise and 1 h before sunset. On the last day of each trial, the bird was weighed (ZIEIS Veterinary Pet Scale; 5 g accuracy) and returned to the main aviary. The length of the birds' tarsus + metatarsus was measured at the end of the entire experiment (Neiko digital caliper; Neiko Tools, Wenzhou, Zhejiang, China; model number: 01409 A; ±0.03 mm accuracy). Three infrared camcorders (Night Owl CAM-600) connected to a DVR (Night Owl Apollo-45 or LTE-44500) continuously recorded the area within the experimental cage and immediately outside (2.5 m from the cage perimeter) the experimental cage.
We determined the number of head movements the birds made (see algorithm below) during each night of the trials (starting 1 h after sunset and ending 1 h before sunrise; "nighttime period"). Using the video recordings, we also calculated the percentage of time that birds spent on the roost during the nighttime periods, the percentage of time that potential predators and non-predators were visible along the perimeter of the experimental cage, when the birds ascended to the roost for the night, and when the birds descended from the roost in the morning. The time at which a bird ascended to the roost for the night was determined by moving backwards in the videos from the nighttime period (1 h after sunset) and finding the time when the bird jumped on the roost. If the bird was not already on the roost 1 h after sunset, then we moved forward in the videos until the bird jumped on the roost. The time at which a bird descended from the roost for the night was determined in a similar manner except that we moved forward in the videos from the nighttime period (1 h before sunrise) until finding the time when the bird jumped off the roost. If the bird was already off the roost 1 h before sunrise, we moved backward in the videos until the bird jumped off the roost. We excluded times when the experimenters interfered with when the bird ascended to the roost or descended from the roost (e.g., if the bird descended from the roost because the experimenter entered the enclosure).

## Head movement extraction

In order to classify head movements using an accelerometer, we needed to examine the accelerometer data with respect to the birds' behavior. Using similar steps as described above, we performed 10 trials in which we video recorded the birds' behavior (Sony SR47) while they were wearing an accelerometer at night (no artificial light was turned on). These trials were performed from April through August 2013 in Durham, NC, USA (36.01°N, 79.02°W) using the same captive population as above (the birds were relocated from North Carolina to Indiana in August 2013).

We synchronized the accelerometer data with the behavioral videos (Logger Pro, Vernier Software and Technology, LLC; Fig. 2; Video S1). We labeled the accelerometer data to indicate when a head movement began and ended. We labeled small head movements (less than 5 deg) and large head movements (greater than 5 deg). The small head movements primarily occurred when the bird blinked or moved its head slightly while sleeping; it is unlikely that these small head movements were related to vigilance behavior and it was necessary to exclude them from the analysis. In order to quantitatively distinguish between small and large head movements, we determined the absolute value of the range of the acceleration in the $x$, $y$, and $z$ and then summed these three ranges ('acceleration range') for each head movement. We found that 70% of the small head movements had an acceleration range below 4.61 m s$^{-2}$ and 70% of large head movements had an acceleration range above 5.30 m s$^{-2}$. We therefore reclassified the coded data such that only head movements with an acceleration range greater than 4.90 m s$^{-2}$ were classified as head movements (Video S2).

We created a custom algorithm (Matlab R2014a; The Mathworks Inc., Natick, MA, USA) to extract head movements from the accelerometer data and used the labeled

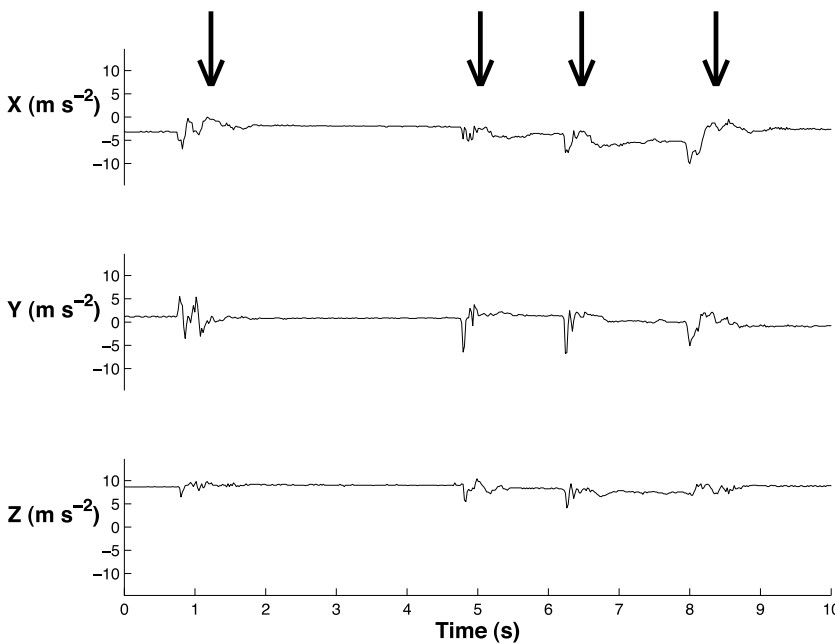

**Figure 2 Sample of the accelerometer data in swing (*X*), sway (*Y*), and yaw (*Z*).** Arrows indicate the four times when the peahen makes a head movement. This graph is also displayed in Video S1.

accelerometer data to examine its accuracy. This algorithm is similar to that used in another study that extracted head movements from accelerometer data (*Kokubun et al., 2011*) because it also relies on a threshold system. Our algorithm performed two steps to extract head movements. First, it identified times at which the change in sway acceleration (delta y) exceeded $1.37$ m s$^{-2}$. This threshold value was determined based on one randomly-selected bird from the labeled dataset. We adjusted this threshold value until the number of predicted head movements most closely matched the number of actual head movements. Second, the algorithm filtered these times to ensure that the same head movement was not counted as multiple head movements. Based on the labeled data, head movements were at least 0.5 s apart. Therefore, this filter removed a head movement if it was within 0.5 s of another head movement.

## Accelerometer effect

We conducted eight trials (with eight different peahens) to test the effect of the accelerometer on the birds' vigilance behavior. These trials were performed in February and March 2013 with the population in Durham, NC, USA. On one night, the bird had an accelerometer attached to its head; on the other night, the bird did not have an accelerometer attached to its head (the order of accelerometer attachment was randomized across trials). The artificial light was not turned on. Two infrared camcorders (Night Owl CAM-600) connected to a DVR (Night Owl Apollo-45 or LTE-44500) continuously recorded the bird. We randomly selected three 10-min periods from both nights of each trial (the times were matched in each night) and manually scored the number of head movements in each period.
## Sleep effect

We conducted eight trials (with eight different peahens) to examine the relationship between head movement rate and sleep behavior. These trials were performed in March and April 2015 with the population in West Lafayette, Indiana, USA in the experimental cage from the artificial light experiment. Each bird had an accelerometer attached to her head and was tested during one night. The artificial light was not turned on. Two infrared camcorders (Bolide Technology Group IR Bullet Camera) connected to a DVR (Swann DVR4-2600) continuously recorded each bird such that the left and right eye of the bird were visible. We randomly choose two 30-min periods (occurring after the bird ascended to the roost for the night and before the bird descended from the roost in the morning) from each trial. For the left and right eye separately, we scored the times at which the eyes were closed (excluding blinks; using Inqscribe software). We scored the left and right eye separately because peahens (*Yorzinski & Platt, 2012*), like other birds (*Rattenborg, Amlaner & Lima, 2000*), asymmetrically close their eyes during sleep. We then determined the percentage of time that both eyes were simultaneously closed ('sleep behavior'); the percentage of time that both eyes were simultaneously closed was strongly correlated with the percentage of time that the right eye was closed ($F_{1,14} = 2{,}168$, $p < 0.0001$, $R^2 = 0.99$) and the left eye was closed ($F_{1,14} = 2{,}683$, $p < 0.0001$, $R^2 = 0.99$).

## Roost selection

We conducted eight trials (with eight different peahens) to examine whether peahens prefer to roost under artificial night lighting ('direct light') or away from the lighting ('low light'). These trials were performed in April and May 2015 with the population in West Lafayette, Indiana, USA in an experimental cage (4.5 m × 9.0 m) that was 75 m from the large aviary. There were two wooden sawhorse roosts (0.85 m tall and 1.3 m long) that were positioned on opposite sides of the cage (1.1 m from the cage sides and 6.8 m from each other). An LED flood light (Philips 17-Watt Outdoor and Security Bright White; model: PAR38) was suspended from the roof directly above each roost (1.8 m from the ground). One of the lights was turned on during each trial (randomized across trials). At night when the light was turned on, the light intensity was 3.0 kLux directly below the light (light meter on roost facing up at light) and 0.22 lux at the roost on the opposite side of the cage (light meter facing toward the light; Extech EasyView 31 light meter; measurements taken during a night with clear skies and 78.0% moon illumination). Two infrared camcorders (Night Owl CAM-600) connected to a DVR (Swann DVR4-2600) continuously recorded each roost. Based on the video recordings, we determined whether the bird spent the night on the roost that was under 'direct light' or 'low light.'

We performed another roost choice experiment to assess whether peahens prefer to roost without any artificial light ('no light') or to roost with low levels of artificial light ('low light'). We tested 16 different peahens in individual trials that each lasted two nights. The trials lasted two nights so that we could determine whether peahens were consistent in their roosting preferences. This experiment was conducted from May to July 2015 in the same cage that was used for the roost choice experiment above. Black plastic divided the

cage in half (lengthwise) but a small opening (0.75 m) did not have black plastic so that the bird could move between the two sides of the cage. The black plastic ensured that light from one side of the cage did not enter into the other side. There was a wooden sawhorse roost (0.85 m tall and 1.3 m long) on both sides of the cage (2 m from the cage side). An LED flood light (Philips 17-Watt Outdoor and Security Bright White; model: PAR38) was suspended from the roof and positioned 4.5 m from each roost (1.8 m from the ground). One of the lights was turned on during each trial (randomized across trials). At night when the light was turned on, the light intensity was 1,260 lux below the light (light meter on ground facing up at light), 0.75 lux at the roost that was in the same side of the cage (light meter facing toward the light), and 0.01 lux at the roost that was in the opposite side of the cage; when the light was turned off, the light intensity was 0.01 lux below the light and 0.01 lux at each roost (Extech EasyView 31 light meter; measurements taken during a night with clear skies and 26.4% moon illumination). Two infrared camcorders (Night Owl CAM-600) connected to a DVR (Swann DVR4-2600) continuously recorded each roost. Based on the video recordings, we determined whether the bird spent the night on the roost that was under 'no light' or 'low light.'

## Data analysis

We tested whether nocturnal vigilance (measured using the number of head movements) varied with respect to lighting. We ran a repeated-measures mixed linear model (PROC Mixed with a variance components (VC) covariance structure) with head movement rate (natural log transformed to meet underlying assumptions of normality) as the dependent variable. The head movement rate was calculated by summing the number of head movements that occurred in the nighttime period and then dividing that sum by the total time in that nighttime period for each night of each trial.

The independent variables were the trial type (light trial or control trial), trial night (the specific night of the trial: 1–7), and their interaction as well as environmental variables (wind speed, precipitation, temperature, moon illumination, predator presence, and non-predator presence) and morphological measurements of the bird (mass and tarsus + metatarsus). The climate variables were obtained from a nearby weather station (http://iclimate.org; ACRE- West Lafayette) and moon illumination was the fraction of the moon's surface that was illuminated from the sun's rays (http://www.timeanddate.com; Lafayette, IN). The wind speed (natural log transformed to meet underlying assumptions of normality) and temperature were averaged across the nighttime period. Since there was no precipitation during 79% of trial nights, precipitation was categorized as being present or not. Predator and non-predator presence was whether predators or non-predators, respectively, were visible along the outside of the perimeter or not during the nighttime period (predators and non-predators were visible in only 34.5% of nights). We performed *a priori* contrasts to test whether head movement rates during each of the seven trial nights differed between the light trials and control trials as well as whether head movement rates differed between night 2 (first night of light) and 5 (last night of light) of the light trials.

We ran two repeated-measures mixed linear models to determine the variables influencing the time (relative to sunset and sunrise) at which the birds ascended to the roost and descended from the roost for the night. The independent variables were the trial type (light trial or control trial), trial night (the specific night of the trial: 1–7), and their interaction as well as environmental variables during the nighttime period (wind speed, precipitation, temperature, and moon illumination) and morphological measurements of the bird (mass and tarsus + metatarsus). We also ran repeated-measures mixed linear models to evaluate whether head movement rate (natural log transformed) (1) differed depending on whether the bird was wearing an accelerometer or not and (2) was related to sleep behavior. We performed binomial tests (Proc Freq) to assess peahens' roosting preferences (the peahens never switched to a different roost during a given night). All analyses were performed in SAS (9.3; Cary, NC, USA) or Minitab (15.1; Minitab Inc., State College, PA, USA). The data supporting this article are available in Harvard Dataverse: 10.7910/DVN/J3RF1P.

## RESULTS

The extraction algorithm accurately predicted the head movements of peahens from the accelerometer data (Fig. 2). Across all the birds, there were 1,699 head movements observed in the labeled dataset and the algorithm predicted that there were 1,678 head movements (overall accuracy: 98.8% correct). Averaging within birds, the overall accuracy was 96.1% (SE: 1.5%). Of the 1,678 head movements that the algorithm predicted, 1,536 of those head movements were true head movements (the predicted head movement fell within the time period of an observed head movement; "true accuracy": 90.4% correct). Averaging within birds, the true accuracy was 87.4% (SE: 3.4%). The accuracies were similar even when excluding the trial from the bird that was used to create the threshold value (see "Materials and Methods"; overall accuracy: 98.8%; true accuracy: 90.6%). The accelerometer did not have an effect on the number of head movements peahens made ($F_{1,7} = 0.15$, $p = 0.71$; Fig. 3). Peahens that had lower head movement rates spent more time sleeping ($F_{1,7} = 31.05$, $p = 0.0008$; Fig. 4).

Head movement rate was related to the trial type (light trial or control trial; $F_{1,22} = 30.45$, $p < 0.0001$), trial night (the specific night of the trial; $F_{6,102} = 7.21$, $p < 0.0001$), and their interaction ($F_{6,102} = 4.67$, p=0.0003). Birds that weighed less had higher head movement rates than birds that weighed more ($F_{1,22} = 13.11$, $p = 0.0015$) but the tarsus + metatarsus length was unrelated to head movement rates ($F_{1,22} = 0.01$, $p = 0.92$). The climate variables and moon illumination had no impact on head movement rate (wind: $F_{1,102} = 2.97$, $p = 0.088$; precipitation: $F_{1,19} = 1.61$, $p = 0.22$, temperature: $F_{1,102} = 1.59$, $p = 0.21$, moon illumination: $F_{1,102} = 0.40$, $p = 0.53$). Importantly, the head movement rates were unrelated to predator and non-predator presence (predator presence: $F_{1,13} = 1.15$, $p = 0.30$, non-predator presence: $F_{1,15} = 0.59$, $p = 0.46$). This is not unexpected given that predator and non-predator presence was rare and these predators and non-predators were outside the cage (and therefore largely visually blocked by the black plastic which surrounded the cage) and not directly under the artificial light.

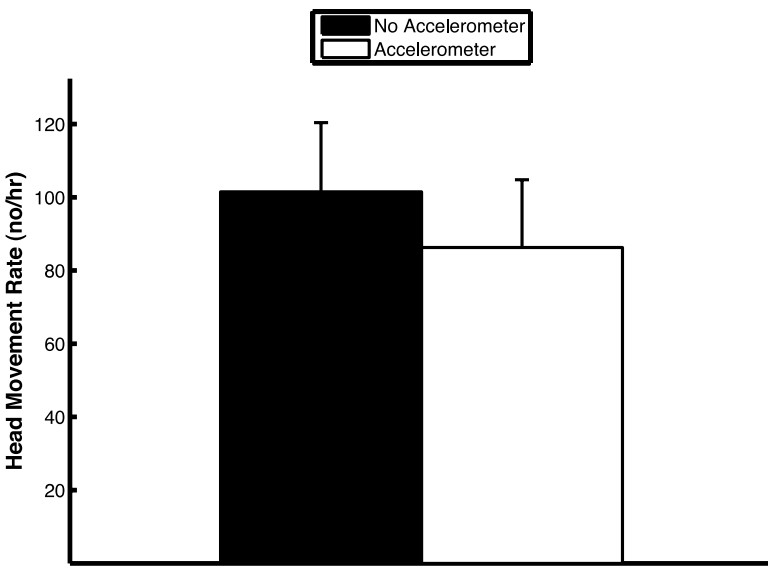

**Figure 3** Head movement rate was similar regardless of whether the peahen was wearing an accelerometer or not (means ± SE).

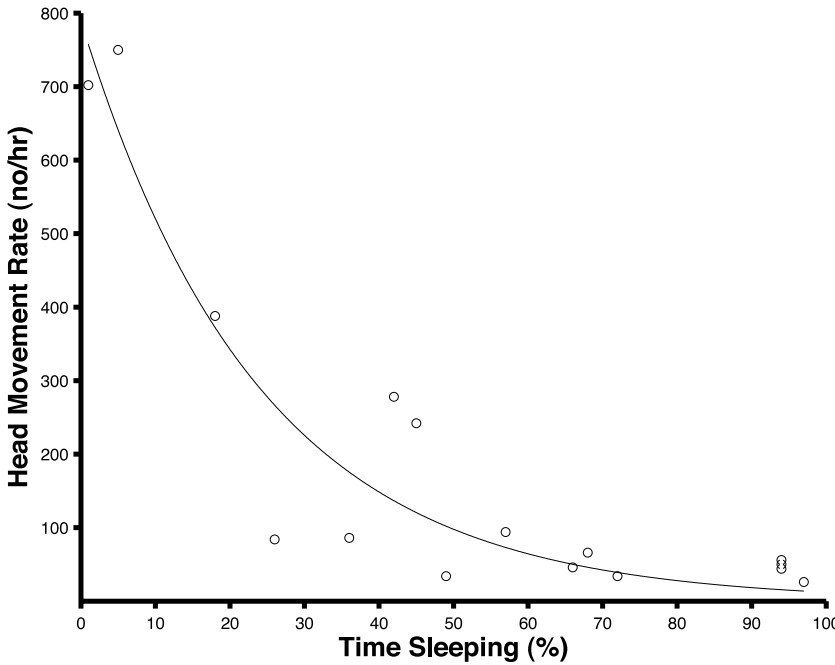

**Figure 4 Peahens that exhibited lower head movement rates spent more time sleeping.** Because each peahen was sampled during two periods (see "Methods"), there are two circles per bird.

However, head movements in peahens are related to antipredator behavior. By manually analyzing head movements from a previous experiment in which peahens were exposed to a taxidermy raccoon at night (without any artificial light pollution; *Yorzinski & Platt, 2012*), peahens made more head movements during a 1-min period while the predator

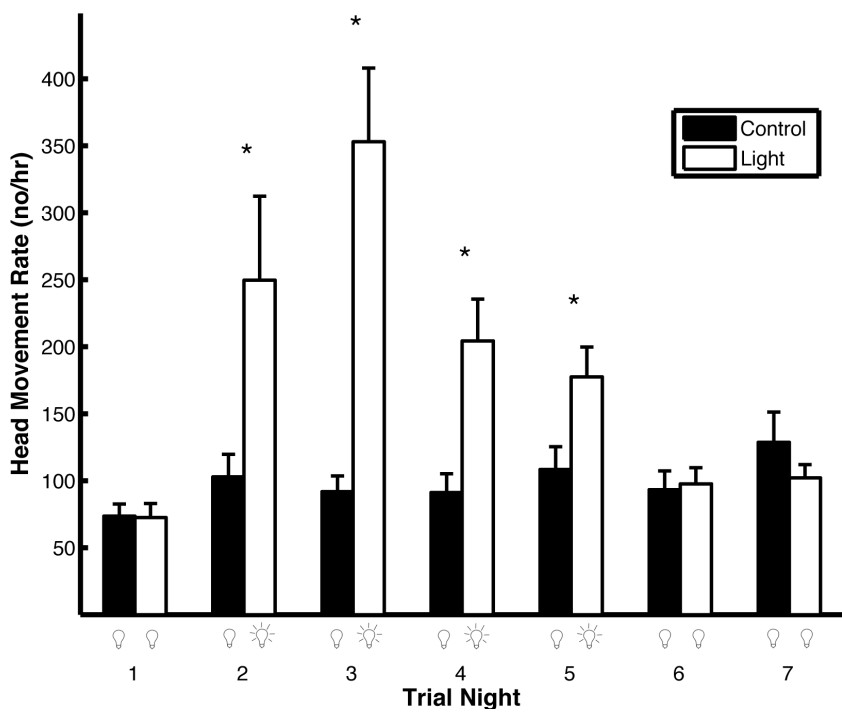

**Figure 5 Artificial light pollution increases head movement rates (means ± SE).** Head movement rates were similar on nights when the artificial light was off in both light and control trials (nights 1, 6, and 7). Head movement rates were significantly higher during nights when the artificial light was on during the light trials and off during the control trials (nights 2–5). Asterisks indicate significant differences in head movement rates between the light and control trials.

was moving toward them and then stopped in front of them (mean ± SE: 6.21 ± 4.14) compared to a 1-min period immediately before the predator was exposed (mean ± SE: 0.80 ± 0.91; paired $t$-test: $n = 7$; $t = 3.77$; $p = 0.009$; we averaged the head movements from the two peahens that were tested in each trial).

Artificial light pollution had a strong effect on head movement rates (Fig. 5). The head movement rate was similar on the first night of both trial types when no light was on ($t_{1,102} = 0.39$, $p = 0.69$). On the second, third, fourth, and fifth nights of the trials, when the light was on during the light trials and off during the control trials, the head movement rate was higher in the light trials compared to the control trials (second night: $t_{1,102} = 5.16$, $p < 0.0001$; third night: $t_{1,102} = 4.28$, $p = 0.0002$; fourth night: $t_{1,102} = 3.52$, $p = 0.0006$; fifth night: $t_{1,102} = 2.13$, $p = 0.036$). On the sixth and seventh nights, when the light was off in both trial types, there was no difference in head movement rate (sixth night: $t_{1,102} = 0.25$, $p = 0.80$; seventh night: $t_{1,102} = 0.23$, $p = 0.82$). During light trials, the head movement rate was higher on the first night that the light was on (night 2) compared to the last night that the light was on (night 5; $t_{1,102} = 2.51$, $p = 0.014$). The results were qualitatively the same if the head movement rate was not log transformed except there was no significant difference between head movement rates during night five in both the light and control trials ($t_{1,102} = 1.68$, $p = 0.096$). If the $p$-values are corrected for multiple comparisons using the Holm–Bonferroni method, there is no significant

difference between head movement rates during night five in both the light and control trials nor between the first night that the light was on compared to the last night that the light was on in the light trials.

Peahens remained on the roost for most (97.2%) of the total nighttime period (the nighttime periods from all the trial nights across both treatments). They roosted on the ground for the entire nighttime period in only 2.3% of trial nights. During trials when they remained off the roost for only a portion of the nighttime period (11 nights), they primarily did so during light trials on nights when the light was on (10 nights). Potential predators (cats, raccoons, opossums, and owls) spent little time (0.25% of the total nighttime period) directly outside the cage. The percentage of time that predators were present outside the cage was unaffected by whether the light was on or off (Kruskal-Wallis: $H = 0.06$; $p = 0.81$). Non-predators (frogs, mice, rabbits, and skunks) spent slightly more time outside the cage (2.36% of the total nighttime period) than predators and they spent more time outside the cage when the light was on compared to when it was off (Kruskal–Wallis: $H = 7.52$; $p = 0.0061$). Peahens ascended to the roost later in the night when the temperature was higher ($F_{1,105} = 4.45$, $p = 0.037$); the other independent variables, including the trial type, did not affect when the birds ascended to the roost ($p > 0.07$). Peahens descended from the roost later in the morning when the moon illumination was higher ($F_{1,109} = 10.12$, $p = 0.0019$); the other independent variables, including the trial type, did not affect when the birds descended from the roost ($p > 0.08$).

Peahens exhibited a strong preference for roosting away from direct artificial lighting ($p = 0.0078$, two-tailed binomial test). In fact, all of the peahens ($n = 8$) chose to roost in 'low light' compared to 'direct light.' In contrast, peahens ($n = 16$) did not show a preference for roosting in 'no light' versus 'low light' conditions (night one: $p = 0.32$, two-tailed binomial test; night two: $p = 0.62$, two-tailed binomial test). Most of the birds (69%) roosted in the same location during both nights of their trials. However, one bird roosted in the dark during the first night and in the low light for the second night while four birds did the opposite.

## DISCUSSION

Artificial light pollution increases nocturnal vigilance in peahens. Peahens exhibited a higher rate of head movements (a proxy of vigilance; *Jones, Krebs & Whittingham, 2007*) on nights when artificial light was present compared to nights when artificial light was absent. These higher head movement rates were not caused by actual threats in the environment—predator presence was rare and unrelated to the number of head movements that peahens made. Furthermore, peahens that exhibited higher head movement rates spent less time sleeping.

Even though animals are increasingly confronted with artificial light pollution, we are only beginning to understand the effects it has on their behavior. Artificial night lighting affects general activity patterns. This is unsurprising given that light is an important factor in mediating circadian rhythms (*Fonken & Nelson, 2014*). Some birds extend the times during which they forage when exposed to artificial light. Mockingbirds feed their nestlings

late in the evening when under high artificial light levels (*Stracey, Wynn & Robinson, 2014*). European blackbirds continue foraging longer into the evening (*Russ, Rüger & Klenke, 2015*) and begin their mornings earlier (*Dominoni et al., 2014*) when exposed to artificial night lighting. Artificial lighting can therefore alter basic activity patterns but the fitness consequences of these changes are unknown. Artificial lighting can even affect physiological processes. Siberian hamsters have reduced immune function when exposed to artificial lighting (*Bedrosian et al., 2011*) and the reproductive systems of birds change under artificial lighting (*Dominoni, Quetting & Partecke, 2013*). During the rare occasions when peahens descended from the roost during the night in this study, they primarily did so during nights when the artificial light was on and they would begin foraging on the ground. Mice also alter their feeding habits when exposed to increased nocturnal lighting and this can lead to excess weight gain (*Fonken et al., 2010*). However, unlike some species (*Dominoni et al., 2014*), artificial lighting did not influence the timing of when peahens ascended to the roost or descended from the roost in the evening or morning, respectively. Because the peahens had unlimited access to food in this captive study, it may have been unnecessary for them to take advantage of increased lighting by maximizing their foraging time.

Artificial light pollution affects predator–prey relationships. Predators, including harbor seals and bats, are more successful at capturing their prey when artificial light pollution is present than absent (*Rydell, 1992*; *Yurk & Trites, 2000*; *Minnaar et al., 2014*). Avian and aquatic predators may also be more successful at capturing prey under artificial night lighting (reviewed in *Frank, 1988*; *Becker et al., 2013*). In response to high predation rates under artificial light, prey can alter their anti-predator strategies. Frogs decrease their calling rates when exposed to artificial nocturnal light and this may reduce their predation risk (*Baker & Richardson, 2006*). This study demonstrates that peahens increase their vigilance rate in response to artificial night lighting.

Vigilance is a key component to understanding the evolution of antipredator behavior (*Caro, 2005*). Individuals that are more vigilant are faster at detecting predators (*Lima & Bednekoff, 1999*). Antipredator vigilance occurs when animals scan their environment for potential predators (*Bednekoff & Lima, 2002*). Head movements are one way in which animals can remain vigilant because it allows them to rapidly shift their visual field (reviewed in *Jones, Krebs & Whittingham, 2007*). Such vigilance can be useful to detect both predators and monitor conspecifics (*Lung & Childress, 2007*). Individuals can also remain vigilant by moving their eyes (*Yorzinski & Platt, 2014*) and "peeking" (periodically opening their eyes while sleeping; *Lendrem, 1984*). Individuals in large groups are often less vigilant than those in small groups (*Lima, 1995*). Vigilance is also affected by where animals choose to sleep. Animals can select sleeping sites with varying levels of vegetation and accessibility to reduce predation risk (*Lazarus & Symonds, 1992*). Some species may prefer roosting under artificial lighting because they can detect predators more easily (*Gorenzel & Salmon, 1995*). In contrast, other prey species may be more vulnerable to predation by sleeping under artificial lighting (*Longcore & Rich, 2007*). In this study, peahens preferred to roost further away from high levels of artificial lighting (although they showed no preference between roosting under low level artificial lighting and no artificial lighting). However,

when the peahens' only option was to sleep near artificial lighting, they exhibited higher vigilance rates than they did when exposed to natural night lighting. Therefore, they may be compensating for increased predation risk by increasing their vigilance levels. Peahens may exhibit low vigilance rates under natural conditions at night (i.e., only moonlight) because they see poorly in low-light environments (*Hart, 2002*; *Yorzinski & Platt, 2012*). It would be informative to present predators to the birds at night to assess their predator detection abilities. Given their increased vigilance levels during nights with artificial light pollution, we would expect them to detect predators more quickly than during nights without artificial light pollution.

We also found that vigilance behavior and sleep are inversely related. Peahens that were more vigilant spent less time sleeping (see also *Gauthier-Clerc, Tamisier & Cézilly, 2000*). We defined sleep as when both eyes of the birds were closed. Measuring their sleep using an electroencephalogram would provide additional information about their sleep stages (*Campbell & Tobler, 1984*). The trade-off between vigilance behavior and sleep may explain why peahens showed decreased vigilance behavior after continued exposure to artificial lighting (their vigilance rate was higher on the first night that the artificial light was present compared to the last night that the light was present). Peahens that maintain high nocturnal vigilance rates may suffer cognitive impairments (*Thomas et al., 2000*) or other costs that outweigh the benefits of being more alert at night.

It can be difficult to obtain accurate measurements of vigilance because animals are frequently engaging in vigilance behavior throughout the day and night. Previous studies generally measure vigilance by manually recording this behavior during a relatively short time-period (e.g., *Jones, Krebs & Whittingham, 2007*). We developed a novel technique to automatically quantify vigilance by using an accelerometer. An accelerometer positioned on the head of an animal can track all of the animal's head movements. This technique is especially powerful for recording nocturnal head movements in diurnal animals because the animals are primarily still at night except for head movements (and the accelerometer will therefore not mistake other behaviors with head movements). It can be a useful tool for future comparative studies to examine the factors, both natural and anthropogenic, that influence vigilance behavior.

## ACKNOWLEDGEMENTS

We thank the Purdue Department of Forestry and Natural Resources, especially Brian Beheler, Ryan Hensley, Matt Kraushar, Michael Loesch-Fries, and Burk Thompson, for allowing us to house the birds on their property and providing logistical support. Kailey Chema, Connor Egyhazi, Fred Hermann, and Diamond Jones helped run some of the trials. Carlo Catoni and Marco Scialotti provided technical support for the accelerometers and John Melville assisted us in using Logger Pro. Merijn DeBakker offered advice in analyzing the accelerometer data. Michael Platt, Barny Dunning, and Esteban Fernández-Juricic provided logistical support.

### Funding

This research was partly funded by Gregg, June, and Vickie Stilwell. The funders had no role in study design, data collection and analysis, decision to publish, or preparation of the manuscript.

### Competing Interests

The authors declare there are no competing interests.

### Author Contributions

- Jessica L. Yorzinski conceived and designed the experiments, performed the experiments, analyzed the data, contributed reagents/materials/analysis tools, wrote the paper, prepared figures and/or tables, reviewed drafts of the paper.
- Sarah Chisholm analyzed the data, reviewed drafts of the paper.
- Sydney D Byerley, Jeanee R. Coy, Aisyah Aziz, Jamie A. Wolf and Amanda C. Gnerlich performed the experiments, reviewed drafts of the paper.

### Animal Ethics

The following information was supplied relating to ethical approvals (i.e., approving body and any reference numbers):

The study was approved by Duke University Animal Care and Use Committee (A205) and Purdue University Animal Care and Use Committee (1305000862 & 1504001232).

### Data Availability

The following information was supplied regarding the deposition of related data:

The data supporting this article are available in Harvard Dataverse: 10.7910/DVN/J3RF1P.

### Supplemental Information

Supplemental information for this article can be found online at http://dx.doi.org/10.7717/peerj.1174#supplemental-information.

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
