# Peer review of "Artificial light pollution increases nocturnal vigilance in peahens"

_PeerJ, doi:10.7717/peerj.1174_

## Round 0.1 · original submission · Minor Revisions

Both reviewers found this an interesting, well designed, and well presented study. Both recommended returning it for minor revisions, making a number of useful suggestions for improved clarity. I concur with their recommendation.

I consider the suggestion of Reviewer 1 for an examination of immune function to be a suggestion for additional research and not a requirement for publishing this manuscript.

In the comments of Reviewer 2, I believe that the references to information 'below' in the final 'Comments for the Author' section refer to material presented the previous sections. It appears that the paragraph was intended for the beginning of the review but was somehow displaced to the end.

Even though Reviewer 2 consider that the manuscript needed only minor revisions, he/she did suggest two additional analyses that could improve the interpretation. First, it is possible that increased vigilance might occur in close temporal proximity to the appearance of potential predators and/or non-predators and that this increase would not be detected in the average vigilance for a whole night. Examining this suggestion would require quantifying vigilance rates during and shortly after the times that the other species were actually present. Second, it is possible that nocturnal illumination reduces sleep and that higher vigilance is an indirect consequence of less time spent sleeping. Although you report a negative correlation between vigilance and sleep, this does not address the question as well as evaluating vigilance rates per time awake would do. It might also be possible to gain some insights by comparing the relative magnitudes of the decrease in sleep and the increase in vigilance While these suggested additional analyses would be likely to add to the overall contribution, I realize that they may not be feasible and do not consider them an absolute requirement for publication. At a minimum, however, the implications for interpreting the results should be included as topics in the Discussion of a revised manuscript.

The following additional comments come from my review of the manuscript should be taken as suggestions with the same weight as reviewers' (rather than editor's) comments:

L29. As with the title, 'birds' is too general a category of investigation in the Abstract.
L32. Be more explicit that you operationally define vigilance as substantial head movement.
L68, 79. 'predate' is a bit awkward because it can mean exist at an earlier date. I prefer 'prey on'.
L74. You go from the topic of lack of knowledge concerning the effects of light pollution on vigilance, to the specific goals of your study. I think that a brief review of what is known about light levels and vigilance and about nocturnal vigilance in general (e.g., its relationship to sleep, security of sleep location, natural variation in light levels) would be appropriate here.
L75. I think that a sentence on the natural distribution of peafowl would make your statement about their occurrence in urban environments clearer. I initially assumed you were referring to the naturalized populations that occur in various places around the world. But the predator references suggest you are referring to their native distribution.
L133. I don't understand the reference to calipers with 60 mm accuracy. They must be much more accurate than that?
L147, 152 and elsewhere. The expression 'descended the roost' seems awkward to me. I would have thought a preposition was required: 'descended from the roost'. 'Ascended the roost' did not initially seem problematic, but it probably should be 'ascended to the roost'.
L171, 172, 180. I was confused to see acceleration measured in grams (g in SI units). An internet search confirmed that the SI unit for acceleration is m/s2 but that it is popularly presented as g for g-force. To avoid reader confusion, I suggest that you either convert to SI units or explain the terminology and not use the abbreviation g, which is well established as indicating grams. (This will also affect your figures/videos when you follow Reviewer 1's request to add units.)
L176. extract not exact
L185, 186 (and elsewhere?). SI unit for seconds is s.
L210. There can be issues about defining sleep without an electroencephalogram. I suggest that you provide some justification for using this definition of sleep.
L321. Lack of a significant difference is not equivalent to being the same.

Reviewer 1 ·

Basic reporting

The title should be specific to peahens, not to birds in general as there are predatory bird species that may not show the same reaction to artificial light at night.

I think it was a simple but good experiment (I like using accelerometers this unique way), and a well written manuscript. There are, however, a few grammatical errors throughout the manuscript.

Fig 1. Provide labels for x and y axis.

Fig 3. It may be better to show that nights 1, 6 and 7 did not have the artificial light on in the figure itself rather than in the figure legend as at a glance it looks like the effect on vigilance only came about on the 2nd night and disappeared by the 6th.

Experimental design

I very much liked the data analysis section and results that went through the rather complicated statistics very clearly (I see they put a statistics person on the paper).

The authors report illuminance data from their light meter. Further characterization of irradiance (watts/m2) and specific wavelengths of light used would make generalization of the results easier.

Validity of the findings

It seems that the article meets the journal standards. If the authors have any biological fluids/samples (blood, urine, feces…) on which to test humoral immune function, it would be interesting to see whether exposure to artificial light and subsequent sleep deprivation reduced this measure of fitness.

Reviewer 2 ·

Basic reporting

Title: The title might be a little too general in the sense that only one species of birds was tested here. I would recommend putting the name of the species studied here.
Line 73: There was a previous study with wallabies on this very same topic (Biebouw, K., & Blumstein, D. T. (2003). Tammar wallabies (Macropus eugenii) associate safety with higher levels of nocturnal illumination. Ethology Ecology & Evolution, 15, 159-172.). I missed other references here on the effect of low light on vigilance (see a review in Biological Reviews 82: 511-525) and a recent review on behavioural adjustments to artificial illumination (Behavioral Ecology 26: 334-339). Also a recent book on vigilance has a section on vigilance at night (Beauchamp (2015). Animal Vigilance. Academic Press, London). These references would be useful to recast some of the statements in the introduction.
Line 83: We could argue the opposite: because it is difficult to see at night animals would be expected to devote more time to vigilance. In flamingos, as in the wallaby study cited above, birds were more vigilant on darker nights (Ethology 109: 511-520).
Line 94: This can be true as long as animals do not use head movements for other purposes such as monitoring neighbours.

Experimental design

Line 119: To be absolutely clear, the same bird was used for 14 nights, 7 for control trials and 7 for light trials (during which only 4 nights had light). This means there were two trials per bird. Is that correct? This was not really clear. Also did you vary the order of presentation of control and light trials for each bird? Later in the results section, it seems to suggest that different birds were used for the different trial types. Clarify this please.
Line 124: It seems that light levels decreased quite rapidly with distance. Can such low light levels actually help a peahen see better in the area where danger is coming from (outside the cage)?
Line 167: At least in gulls, when the birds are sleeping, they open their eyes periodically without moving the head. This was considered to be vigilance. I agree that the large head movements are probably related to vigilance but I think a quite a sizable amount of vigilance cannot be tracked with head movements.
Line 210: What about the possibility of asymmetric eye closure? This has been documented in a large range of species (see the work by Amlaner and Lima for instance).
Line 267: What is a non-predator presence?
Line 270: So you performed many contrasts if I understand correctly. Did you control for the inflation of type I error rate associated with multiple testing?

Validity of the findings

Line 307: This is an odd result in the sense that if the birds are vigilant to detect predators or at least sources of disturbances they should spot these disturbances. Perhaps it would be necessary to see head movement rate just before and after predator presence instead of measuring overall head movement rate, which probably swamped the temporary effect. I suggest looking at this to examine this hypothesis.
Line 378: Perhaps the real conclusion is that peahens decreased their sleeping with the lights on. To show an effect on vigilance, I would like to see what they are doing when not sleeping. They could just be moving their heads with no real purpose. In the wallaby study, the floodlights provided quite a large amount of light and animals decreased their vigilance. Flamingos were also less alert on bright nights but again moonlight flooded the whole habitat. This should be discussed.

Additional comments

In this experimental study, the effect of artificial lighting on vigilance was studied in peahens. There have been few attempts to investigate vigilance at night and fewer still on the effect of artificial lighting. The authors also use an accelerator to measure head movements thought to be indicative of vigilance, which allowed continuous recording of vigilance over the night. The results showed higher head movement rates on illuminated nights.
This study scores points for investigating vigilance outside the box and using a promising technology to measure vigilance. I think head movement rate has its limits to measure vigilance (blinks can provide useful information about disturbances but would not be recorded). In addition, it is not totally clear that head movements served for vigilance. What is clear is that peahens slept less on illuminated nights. It would be nice to show that head movements at night can actually serve to detect disturbances. I propose below a way of showing this with the data. I have other more minor points listed below.

---

## Round 0.2 · accepted · Accept

I am glad that you found the comments by the reviewers and by me helpful. Following the changes, I consider the manuscript suitable for publication.